# Ouabain Reverts CUS-Induced Disruption of the HPA Axis and Avoids Long-Term Spatial Memory Deficits

**DOI:** 10.3390/biomedicines11041177

**Published:** 2023-04-14

**Authors:** Jacqueline Alves Leite, Ana Maria Orellana, Diana Zukas Andreotti, Amanda Midori Matumoto, Natacha Medeiros de Souza Ports`, Larissa de Sá Lima, Elisa Mitiko Kawamoto, Carolina Demarchi Munhoz, Cristoforo Scavone

**Affiliations:** 1Departament of Pharmacology, Institute of Biomedical Sciences, University of São Paulo, São Paulo 05508-000, Brazil; 2Department of Pharmacology, Institute of Biological Sciences, Universidade Federal de Goiás, Goiânia 74690-900, Brazil

**Keywords:** ouabain, CUS, long-term memory, extinction fear memory, corticosterone

## Abstract

Ouabain (OUA) is a cardiotonic steroid that modulates Na+, K+ -ATPase activity. OUA has been identified as an endogenous substance that is present in human plasma, and it has been shown to be associated with the response to acute stress in both animals and humans. Chronic stress is a major aggravating factor in psychiatric disorders, including depression and anxiety. The present work investigates the effects of the intermittent administration of OUA (1.8 μg/kg) during the chronic unpredictable stress (CUS) protocol in a rat’s central nervous system (CNS). The results suggest that the intermittent OUA treatment reversed CUS-induced HPA axis hyperactivity through a reduction in (i) glucocorticoids levels, (ii) CRH-CRHR1 expression, and by decreasing neuroinflammation with a reduction in iNOS activity, without interfering with the expression of antioxidant enzymes. These changes in both the hypothalamus and hippocampus may reflect in the rapid extinction of aversive memory. The present data demonstrate the ability of OUA to modulate the HPA axis, as well as to revert CUS-induced long-term spatial memory deficits.

## 1. Introduction

Chronic stress is associated with the development of neuropsychiatric disorders, for instance, post-traumatic stress disorder (PTSD) or depression- and anxiety-related disorders, where changes in the hormones and neuropeptides that are related to stress ensue, such as in the production and release of corticotropin-releasing hormone (CRH) [1,2]. This neuropeptide is widely distributed in the central nervous system (CNS) and its expression is stimulated by neurotransmitters, such as serotonin and norepinephrine, and by cytokines, such as the interleukin (IL)-1 and -6 and the tumor necrosis factor (TNF) -α [3,4,5].

CRH initiates the response of the hypothalamic–pituitary–adrenal axis (HPA) at the pituitary level and modulates the brain regions that regulate the behavioral responses to stress. This CRH activity occurs through its G protein-coupled receptors (GPCRs), CRHR1 and CRHR2. The CRHR1′s CRH effect is the main receptor that is responsible for the synthesis and secretion of ACTH regulation, which stimulates the release of glucocorticoids from the adrenal cortex [6]. However, studies have observed that CRH overproduction in mice is directly related to the development of anxiety-like behavior [7,8]. Additionally, CRHR1 and CRHR2 appear to modulate the expression of stressors differently. CRH1 is related to the initial activation of the HPA axis to a stress stimulus and anxiogenic response [9,10]. In contrast, CRHR2 activation mediates the stress adjustment response, promoting anxiolytic and antidepressant effects [6,9].

It has been suggested that endogenous ouabain (OUA) levels are modulated by stress conditions [11], however, little is known about the effect of this cardiosteroid in stress situations. The plasma membrane protein, Na^+^,K^+^-ATPase, functions by maintaining cellular ion homeostasis [12,13]. Several CNS disorders are related to alterations in this Na^+^,K^+^-ATPase activity, such as depression and bipolar disorder [14,15,16]. Furthermore, knowing that CUS promotes neuroinflammatory sensitization [17] and that OUA has anti-inflammatory effects [18], in this work, we aimed to evaluate the role of an intermittent treatment with OUA in chronic unpredictable stress (CUS)-induced HPA axis hyperactivity, and its consequences on some inflammatory cytokines, antioxidant enzymes, long-term memory, and the extinction of fear memory. Our results demonstrated that this intermittent treatment with OUA was effective in reversing the CUS-induced HPA axis hyperactivity by reducing the glucocorticoid circulating levels. OUA prevents CUS-induced long-term memory impairment, as well as promotes the facilitation of fear memory extinction. Moreover, OUA reduces the Crh mRNA levels in the hypothalamus, in addition to reducing the Crh1 and Crh2 expression in the hippocampus. Furthermore, OUA decreases the iNOS activity in the hippocampus by altering the CUS-induced low-grade neuroinflammation. These findings suggest that the participation of the OUA as a regulator of long-term memory in animals that were submitted to unpredictable chronic stress and contextual fear memory, independent of the CUS protocol.

## 2. Materials and Methods

### 2.1. Animal and Chronic Unpredictable Stress (CUS)

Male Wistar rats (*n* = 212; 53 per group) (250–350 g) (Biomedical Sciences Institute, University of São Paulo) were kept under a 12 h light/dark cycle (lights on at 7:00 a.m.) and fed ad libitum. The rats were randomly assigned into four groups: PBS, OUA, CUS, and CUS + OUA, with all of them having an intraperitoneal (i.p.) administration of either a vehicle (PBS) or ouabain (1.8 μg/kg) [19] (Sigma-Aldrich, St. Louis, MO, USA) one hour before the stress protocol, every other day. The animals from the CUS and OUA + CUS groups were submitted to different stressor stimuli for 14 days, which was performed as follows: day 1 (2:00 p.m.), restraint, 60 min; day 2 (9:00 a.m.), forced swim, 15 min; day 3 (3:00 p.m.), cold isolation, 90 min; day 4 (7:00 p.m.), lights on, overnight; day 5 (10:00 a.m.), forced swim, 5 min; day 6 (7:00 p.m.), water and food deprivation, overnight; day 7 (2:00 p.m.), restraint, 120 min; day 8 (3:00 p.m.), lights off, 120 min; day 9 (9:00 a.m.), forced swim, 5 min; day 10 (7:00 p.m.), lights on, overnight; day 11 (2:00 p.m.), cold isolation, 90 min; day 12 (9:00 a.m.), restraint, 60 min; day 13 (7:00 p.m.), water and food deprivation, overnight; and day 14 (9:00 a.m.), restraint, 60 min [17]. (Figure 1A). All the animals were euthanized 24 h after the last stressor protocol. From each animal, trunk blood was collected and centrifuged at 3000 rpm for 10 min to obtain the serum, and the hippocampus and hypothalamus were dissected for biochemical studies. Independent experiments were performed for the behavioral tests. All the procedures were also approved by the Ethical Committee for Animal Research (CEUA) of the Biomedical Sciences Institute of the University of São Paulo.

### 2.2. Measurement of Corticosterone, CRH, ACTH, and Cytokine Levels

ELISA kits were used to measure the serum levels of the corticosterone, CRH, and ACTH, following manufacturer’s instructions of the Corticosterone EIA kit (32–20,000 pg/mL—Enzo Life Sciences, Farmingdale, NY, USA), ACTH EIA kit (0.08–1.78 ng/mL- PHOENIX), and CRF EIA kit (0.33–3.73 ng/mL -PHOENIX). The hippocampal and hypothalamus levels of TNF-α (39.1–2500 pg/mL) and IL-1β (39–5000 pg/mL) were measured by ELISA (eBioscience, San Diego, CA, USA), and the results were normalized by a protein concentration. The absorbances were measured using a spectrophotometer at 450 nm (Epock, Biotech, Bothell, DC, USA), and the concentrations of the cytokines were measured by correlating them with the standard curve.

### 2.3. Measurement of Neuronal (nNOS) and Inducible (iNOS) NOS Activity

The constitutive NOS activity was measured by the [^3^H]L-citrulline assay method, as previously described, with some modifications [20]. This assay is based on the biochemical conversion of L-arginine to L-citrulline by NOS. For the NOS activity assay, the hypothalamus and hippocampus were homogenized in a buffer containing: 20 mM HEPES pH 7.4; 0.32 M sucrose; 0.1 mM EDTA; 1.0 mM DTT; 1.0 mM PMSF; 10 µg/mL leupeptin; and 2 µg/mL aprotinin. Then, the tissue suspension was centrifuged at 1000× *g* for 10 min, and the supernatants were centrifuged again at 12,000× *g* for 20 min, at 4 °C. After determining the protein concentration of the sample with the Bio-Rad kit, the samples were diluted to a concentration of 1 µg/µL, and were then incubated for 30 min at 37 °C in 200 µL reaction medium containing: 20 µM arginine (0.5 μCi [3H]L-arginine), 4 µM FAD, 4 µM FMN, 10 µM BH4, 10 µg/mL Calmodulin, and 1 mM NADPH and 100 µL sample. For the iNOS activity (calcium/calmodulin-free activity), EDTA/EGTA were added, and CaCl2 and calmodulin were omitted. The nNOS activity was calculated as the difference between the calcium-calmodulim sample and the EDTA/EGTA sample. To stop the reaction, the tubes were placed on ice, and 1 mL of 20 mM HEPES pH 5.5 was added. The total volume that was contained in the reaction portion was transferred to a column Dowex 50WX8-400, sodium form, to remove the excess of the substrate. The resin was left to settle for 30 min at room temperature, the supernatant was carefully removed in vials with scintillation liquid, and the radioactivity of the L-[3H] citrulline was quantified. The results were normalized by a protein concentration and the NO synthase activity was expressed as pmol/mg min [20].

### 2.4. Real-Time PCR

The total tissue RNA (hypothalamus and hippocampus) was found and purified using the total KIT RNA I (OMEGA, Norcross, GA, USA). The RNA was quantified, and 1μg was treated with DNAse I and subjected to a reverse transcription using oligo (12–18), random primer, and an IMPROM II reverse transcriptase, according to the manufacturer’s instructions (Promega, Wisconsin, WI, USA). The Crh, Crhr1, Crhr2, Sod1, Sod2, and Gsr gene expressions were measured by a quantitative PCR (qPCR), using the TaqMan gene expression assay (Thermo Fisher Scientific, Waltham, MA, USA).

The qPCR reaction was performed in duplicate in the 7500 Fast Real-Time PCR System (Applied Biosystems, Waltham, MA, USA). Each duplicate reaction contained 4 μL of cDNA, 6.25 μL of TaqMan FAST Advanced Master Mix (Applied Biosystems), 0.625 μL of the TaqMan probe, and 1.62 μL of water nuclease-free (Applied Biosystems), totaling a volume of 12.5 μL per well. The first step of the reaction was to amplify at 95 °C for 20 s, followed by 40 cycles at 95° for 3 s (denaturation), and 60 °C at 30 s (annealing and extension). The comparative method of delta-delta-Ct was used to quantify the difference between the samples that were normalized by the calibrator of the endogenous control. As the endogenous control of the qPCR reaction, the TaqMan probe for the Hprt-1 gene (hypoxanthine phosphoribosyltransferase 1) was used. Information on all the probes that were used is listed in Table 1.

### 2.5. Novel Object Recognition (NOR)

Learning and memory tests were performed in open field apparatus (the height of the apparatus was 45 cm, with a circular floor of 75 cm in diameter) and were divided into three phases: habituation (10 min), training (5 min), and test (5 min). The habituation phase was performed 24 h after the stressor stimulus. For three consecutive days, the rats were placed in the center of the apparatus, where they could explore an open field arena with an absence of any objects for 10 min. The training phase was performed 24 h after the last habituation. In the training phase, two identical objects were placed equidistant from the center of the arena, and equidistant from the arena walls. The rats were placed in the arena with their heads positioned opposite to the two identical objects and allowed to freely explore the objects for 5 min. The testing phases were performed 24 h after the last training, and in the testing phase, the rats were allowed to explore the arena that contained one of the familiar objects and a novel object for 5 min. The objects and the apparatus were cleaned between the trials with a 75% alcohol solution. The test was filmed, and later, the time of exploration for each object was measured with the aid of a stopwatch, which was performed blindly. The animals that had an exploration time of under 10 s were excluded from the experiment. The scaled index was calculated as the difference between the time of exploring the new and the old object, and was exposed as a ratio to the total time spent [21].

### 2.6. Contextual Fear Conditioning and Memory Extinction

In order to assess the context-dependent fear memory and extinction, the different groups of rats were exposed to a fear conditioning protocol and subsequently subjected to a context-dependent extinction protocol. Briefly, 24 h after the last stressor stimulus, the animals were placed individually in the conditioning box, which comprised three white walls, a cover, and a transparent acrylic wall (28 × 26 × 23 cm), as well as a base that was composed of bars (a diameter of 0.4 cm and spacing between them of 1.05 cm) that were connected to an electric shock generator (Insight Equipamentos, Pesquisa e Ensino, Ribeirão Preto, Brazil).

The rats were allowed to freely explore the test box for 2 min before receiving a 1 s foot shock (FS, 1 mA). The rats were removed from the chamber 15 s after this foot shock. The rats were tested 24 h after conditioning for their recall and extinction of context-dependent fear for seven consecutive days. Each extinction session consisted of re-exposing the animals to the conditioning context for 10 min without negative reinforcement. The measure of the fear behavior analysis was the freezing time, which was defined as the complete immobility of the animal, with no movement of vibrissae or sniffing. Furthermore, a group of animals was conducted to the unpaired arena, 24 h after the foot shook, and were the control of the experiment. After each test, the matched and unpaired conditioning box was cleaned with 5% alcohol. A video camera recorded all the training and testing procedures, and a behavior analysis was performed blindly by a human analyzer who counted the freezing time of each animal per group [22].

### 2.7. Statistics

The qPCR results were analyzed via the delta-delta-Ct method. The statistical analyses were conducted through a two-way ANOVA, followed by Newman–Keuls post-test. To compare the data between two experimental groups, a t-student test was performed. The differences were significant at *p* < 0.05, and all the results are expressed as the mean ± standard error of the mean (SEM) of the indicated number of experiments. All the analyses were performed using the Prism 6 software package (GraphPad Software, San Diego, CA, USA). The experiments were repeated 2 or 3 times independently, and each experiment had 5 animals per group. The variation in the number of samples was due to the use of the samples for more than one experimental technique, or the loss of samples during processing.

## 3. Results

### 3.1. Ouabain Interferes with HPA Axis Hyperactivity Induced by CUS

Chronic stress induces HPA axis hyperactivity, which is modulated by increased and sustained corticosterone levels [23]. Thus, the effect of a chronic intermittent treatment (every other day) of OUA on the serum corticosterone levels in animals that had been submitted to the CUS protocol was evaluated. It was observed that only the animals that received a chronic intermittent treatment with OUA at the dose of 1.8 μg/kg via intraperitoneal injection did not have altered basal levels of corticosterone (Figure 2A). However, the animals that were submitted to CUS showed an increase in their corticosterone levels 24 h after the last stressor stimulus, when compared with the control group (F (1, 24) = 0.58; *p* = 0.0001 for the CUS factor; *p* < 0.0001 for Ctrl-Ctrl x CUS-Ctrl) (Figure 2A). In this study, the animals that were submitted to CUS+OUA exhibited a reduction in their corticosterone levels when compared with the CUS-only group (F (1, 24) =8.108; *p* = 0.0089 for the treatment factor; *p* < 0.0001 for Ctrl-CUS x CUS-OUA) (Figure 2A). Nonetheless, alterations in the serum CRH and ACTH levels 24 h after the CUS protocol, when compared with the control group (CTR), were not observed (CRH: F (1, 51) = 3.117; *p* = 0.0835 for the CUS factor for Ctrl-for CTR x CUS); (ACTH: F (1, 35) = 1.254; *p* = 0.2704 for the CUS factor for Ctrl-for CTR x CUS). Additionally, the treatment with the OUA did not interfere with the level of CRH (Figure 2B), (F (1, 51) = 0.09387; *p* = 0.7606 for the treatment factor for CUS x CUS+OUA), as well as with the ACTH level [F (1, 35) = 3.376; *p* = 0.0747 for the treatment factor for CUS x CUS+OUA], when compared with the CUS.

### 3.2. Effect of Chronic Unpredictable Stress (CUS) and Ouabain on Inflammatory Parameters in the Hippocampus and Hypothalamus

Based on the evidence that stress can induce inflammatory responses in neurons [24], the TNF-α and IL-1β levels were measured in the hypothalamus and hippocampus. It was observed that the animals that were submitted to CUS did not show differences in their levels of IL-1β in the hypothalamus, when compared with the control (F (1, 16) = 4.914; *p* = 0.0415 for the CUS factor for Ctrl-Ctrl x CUS-Ctrl), and in addition, the same result was observed for the TNF-α (F (1, 34) = 4.338; *p* = 0.0449 for the CUS factor for Ctrl-Ctrl x CUS-Ctrl) (Figure 3A,B). Furthermore, no significant changes were observed between the levels of the IL-1β in the hippocampus of the animals that were submitted to CUS, compared with the control (F (1, 16) = 6.002; *p* = 0.0262 for the CUS factor for Ctrl-Ctrl x CUS-Ctrl), and the same was observed for the levels of TNF-α (F (1, 30) = 0.3623; *p* = 0.5518 for the CUS factor for Ctrl-Ctrl x CUS-Ctrl) (Figure 3C,D). Moreover, the chronic treatment with OUA did not interfere with the levels of IL-1β (F (1, 16) = 0.03912; *p* = 0.8457 for the treatment factor for CUS x OUA-CUS) and TNF-α in the hypothalamus (F (1, 34) = 0.04273; *p* = 0.8375 for treatment factor for CUS x OUA-CUS), when compared with CUS (Figure 3A,B), and a statistically similar result was observed for the IL-1β (F (1, 16) = 0.7027; *p* = 0.3077 for the treatment factor for CUS x OUA-CUS) and TNF-α levels (F (1, 30) = 2.289; *p* = 0.1408 for the treatment factor for CUS x OUA-CUS) in the hippocampus of the animals that were submitted to CUS. Furthermore, no variations in the activity of the total NOS and neuronal isoform (nNOS) were observed in the hypothalamus (Figure 4A,B) and hippocampus (Figure 4D,E) of all the groups. However, there was an increase of the iNOS enzyme activity in the hippocampus of the animals that were submitted to CUS, which was reduced by the ouabain treatment (F (1, 16) = 5.871; *p* = 0.0276 for the treatment factor; *p* < 0.01 for Ctrl-CUS x CUS-OUA) (Figure 4F), even though there were no changes observed in the hypothalamus (Figure 4C).

### 3.3. OUA Did Not Alter the Effects of CUS-Exposure on Antioxidant Enzymes Expression

Chronic stress promotes reactive oxygen species (ROS) generation, causing oxidative stress and, consequently, neurodegeneration. Furthermore, a reduction in the antioxidant enzyme activity was observed in an unpredictable chronic stress model [25,26]. Accordingly, a qPCR was performed to measure the gene transcription levels that were related to the antioxidant enzymes, superoxide dismutase 1 and 2 (SOD1 and SOD2) and glutathione reductase (GSR). Our data demonstrated that, for the 14 days, the CUS protocol did not change the expression of SOD1 (F (1, 34) = 0.4205; *p* = 0.5211 for the CUS factor), SOD2 (F (1, 34) = 0.5226; *p* = 0.4747 for the CUS factor), or GSH (F (1, 32) = 0.3697; *p* = 0.5475 for the CUS factor) in the hypothalamus, when compared with the CTR group. In addition, the results demonstrated that the treatment with OUA did not change the hypothalamic mRNA expression of SOD1 (F (1, 34) = 2.601; *p* = 0.1161 for the treatment factor), SOD2 (F (1, 34) = 1.922; *p* = 0.1746 for the treatment factor), or GSR (F (1, 32) = 6.277; *p* = 0.0175 for the treatment factor), when compared with CUS (Figure 5A–C). Interestingly, the CUS protocol, for 14 days, promoted a reduction in the expression of SOD2 in the hippocampus, when compared with the CTR group (F (1, 34) = 55.82; *p* < 0.0001 for the CUS factor), and this alteration was not prevented by the treatment with OUA (F (1, 34) = 0.01967; *p* = 0.8893 for the treatment factor) (Figure 5E). In addition, we observed that the mRNA GSH levels were reduced in the animals with CUS+OUA, when compared with the OUA group in the hippocampus (Figure 5F). Additionally, no variation in the mRNA levels was detected for the SOD1 in the hippocampus (F (1, 34) = 0.1041; *p* = 0.7489 for the treatment factor).

### 3.4. The Crh, Crhr1, and Crhr2 Gene Expression was Modified by Both CUS and Chronic Intermittent Ouabain Treatment in the Hippocampus and Hypothalamus

The CRH and its receptors are crucial to the regulation process of the HPA axis. Thus, the Chr, Crhr1, and Crhr2 mRNA expression were assayed in the hippocampus and hypothalamus through a qPCR, in order to evaluate the OUA treatment’s effect on the animals that were submitted to CUS. In the hypothalamus, the results demonstrate an increase in the Crh expression of the animals that were submitted to the CUS, in comparison to the control group (CTR) (F (1, 30) = 8.754; *p* = 0.0060 for the CUS factor), and the OUA treatment effect reduced this effect (F (1, 30) = 12.83; *p* = 0.0012 for the treatment factor) (Figure 6A). However, there were no changes in the expression of the Crhr1 in the hypothalamus of the groups that were evaluated (F (1, 34) = 0.08910; *p* = 0.7671 for the CUS factor) and (F (1, 34) = 5.311; *p* = 0.0274 for the treatment factor) (Figure 6B).

Interestingly, we observed that the chronic intermittent OUA treatment promoted an increase in the expression of the Crhr2 in the animals that were submitted to the CUS, when compared with the CUS group (F (1, 32) = 6.005; *p* = 0.0199 for the treatment factor) (Figure 6C). No change in the CRH expression was demonstrated in the hippocampus of the animals after 14 days of CUS, when compared with CTR (F (1, 32) = 8.804; *p* = 0.0056 for the CUS factor) (Figure 6D). When we evaluated the hippocampus, we did not observe changes in the expression of the Crh1 of the animals that were submitted to CUS (F (1, 34) = 5.795; *p* = 0.0216 for the CUS factor) (Figure 6E). Interestingly, we observed a reduction in the Crhr1 expression in the hippocampus of the animals that were treated with OUA and submitted to CUS, when compared with CTR-OUA (F (1, 34) = 0.1837; *p* = 0.6709 for the treatment factor) (Figure 6E). In addition, a reduction in the Crhr2 expression was observed in the hippocampus of the animals that were submitted to CUS and treated with OUA (F (1, 30) = 3.609; *p* = 0.0671 for the treatment factor) (Figure 6F).

### 3.5. CUS-Induced Long-Term Memory Impairment Reduced by OUA

The novel object recognition test revealed that the CUS group exhibited a long-term memory impairment, since they had a reduced capacity to discern the presence of a new object when compared to the control group (F (1, 26) = 0.08254; *p* = 0.7762 for the CUS factor). Furthermore, the chronic intermittent treatment with OUA alone did not interfere with long-term memory formation (Figure 7). Interestingly, the animals that were treated with OUA and were exposed to CUS had a deficit induced by CUS into their long-term memory prevention (F (1, 26) = 6.488; *p* = 0.0171 for the treatment factor) (Figure 7).

### 3.6. OUA Promotes A Rapid Extinction of Fear Memory without Interfering with the Acquisition of Aversive Memory

Finally, OUA and CUS promoted a rapid extinction of fear memory without interfering with the acquisition of aversive memory. The contextual fear conditioning and extinction tests were performed to evaluate the effects of the OUA on the process of the acquisition and extinction of conditioned contextual fear memory. The controls and stressed animals, either treated with chronic intermittent OUA or saline, received a 1 mA foot shock. To evaluate whether the conditioned fear memory was associated with the context of the traumatic event, another group of animals was transferred to a previously unknown container 24 h after receiving the foot shock (unpaired control), and their freezing behavior was analyzed.

These unpaired control animals did not present as much freezing behavior as the ones that were placed in the foot shock context (Appendix A). These results support our data, indicating that the animals were freezing in response to the aversive context (Appendix A). In addition, the animals that were submitted to CUS (F (1, 37) = 0.3469; *p* = 0.5595 for CUS factor), as well as to the chronic intermittent OUA (F (1, 37) = 3.840; *p* = 0.0576 for treatment factor), did not contrast from CTR in their acquisition of fear memory (percentage of freezing time), at 24 h after the acquisition training (day 1) (Figure 8B). Interestingly, regarding the process of memory extinction, the OUA, OUA + CUS, and CUS groups presented a reduction in freezing after successive re-exposures in the contextual fear conditioning arena from the third day (Figure 8B), in comparison with the control group (F (1, 37) = 12.94; *p* = 0.0009 for CUS factor), (F (1, 37) = 4.663; *p* = 0.0374 for the treatment factor). Given this, it is possible to infer that the intermittent treatment with OUA played an important role in the enhancement of aversive memory extinction in non-stressed rats, as did the CUS protocol alone.

## 4. Discussion

Ouabain (OUA), a ligand of Na^+^,K^+^-ATPase, has been identified as an endogenous hormone that is present in human plasma and appears to be involved in the response to acute stress in animals and humans [11]. Moreover, chronic stress is a known important aggravating factor of psychiatric disorders.

The CUS protocol was used to evaluate the role of an intermittent OUA treatment in the modulation of the HPA axis, long-term memory, and the extinction of fear memory, which are parameters that are well known to be altered in depression, anxiety disorders, and PTSD.

The results suggested that a chronic intermittent OUA treatment diminished the CUS-induced HPA axis hyperactivity by reducing the circulating levels of glucocorticoid (Figure 2A). Chronic stress promotes the prolonged activation of the HPA axis, providing long-term adaptive changes in tone and responsiveness and leading to an increased corticosterone secretion, which is important for the modulatory effects of stress on the consolidation and extinction of memory [27,28]. The 14-day CUS protocol leads to a moderate increase in the glucocorticoid levels [29], which were reduced by the chronic intermittent OUA treatment. Previous data have shown that acute OUA does not interfere with the levels of corticosterone that are released in acute stress situations [19], thus suggesting that the OUA that was administered in the chronic intermittent schedule can regulate the activity of the HPA axis, only in response to chronic stressors.

In fact, the OUA intermittent treatment reversed the CUS-induced disruption of the HPA axis, restoring the corticosterone levels. This is probably responsible for the improvement of long-term memory through object recognition that was observed in the CUS group treated with the OUA (Figure 7). Higher levels of corticosterone reduce this object recognition 24 h after training [30].

Despite the 14-day CUS increasing the CRH mRNA expression in the hypothalamus, OUA was able to reduce the CRH to basal levels and increase the CRHR2 mRNA levels in the CUS group (Figure 6). In the hippocampus, the CRHR1 and CRHR2 mRNA expressions were lessened in the CUS group that was treated with OUA (Figure 6). The CRHRs are genes that are involved in stress regulation. An increase in the mRNA levels of CRHR2 in the hippocampus of the mice that were submitted to the stress protocol relates to the worst performance in the memory tests, such as in novel object recognition [31]. Our data show that the treatment with OUA reduced the Crh2 expression in the hippocampus of the rats that were subjected to CUS, as well as an improved performance in the novel object recognition memory test (Figure 7).

It is known that cortisol (corticosterone in animals) and ACTH are secreted in pulses in both humans and rodents [32]. Our data demonstrated that the animals that were submitted to CUS had an increase in their corticosterone levels, however, after 24 h, there were no changes in their ACTH levels (Figure 2). For such an observation, a possible explanation is the ACTH half-life, which is estimated at 22 min after the pulse. A pulse usually has a half-duration of 23 min [33]. Since the serum samples were collected 24 h after the last stressful stimulus, it is difficult to predict the exact time point of a possible pulse.

Another important point that we must highlight is the plasticity of the HPA axis. Previous studies have reported that, in humans, the maintenance of high levels of cortisol in the postoperative period was observed, despite the fact that the ACTH levels quickly returned to baseline levels, which suggests that there is an increase in the adrenal sensitivity to ACTH [34]. This adrenal hypersensitivity can also be explained by the systemic increase of inflammatory cytokines, which also occurs in animals that are challenged with LPS [35,36]. Indeed, chronic stress also promotes a systemic increase in inflammatory cytokines [37,38], which would explain the increase in the corticosterone levels, regardless of the increase in the ACTH levels.

Different studies have reported that OUA is an essential regulator of the inflammatory response in the peripheral and CNS [18,19,39]. Additionally, it is well known that chronic stress promotes, in humans and animals, an inflammatory state in the peripheral and CNS [40,41,42]. Studies have associated these elevated levels of glucocorticoids with the presence of an increased inflammatory response in cells such as macrophages and microglia [43]. Additionally, IL-1β participates in the induction of memory impairment, as well as in the release of CRH [44,45]. However, in the chronic stress model that was performed in this study, it was not possible to observe any changes in the pro-inflammatory cytokines levels in the hypothalamus and hippocampus after 24 h from the last CUS protocol (Figure 3). These findings are in agreement with a previous study that performed CUS for 14 days in rats, where CUS alone did not modulate the expression of the pro-inflammatory cytokines [46]. Nonetheless, an increase in IL-1β and TNF-α in the hippocampus was observed after 1 h of the last CUS protocol, in male mice subjects with 14 days of CUS [47], suggesting that perhaps these cytokines presented a peak immediately after the stress. However, some studies indicate that the priming effects of CUS in neuroinflammatory events pertain to peripheral immunological challenges [17,46]. Furthermore, the participation of the pro-inflammatory cytokines in the development of the neuroinflammation that is induced by chronic stress and the involvement of nitric oxide in the development of anxiety have been demonstrated, through the treatment with L-NAME, a NOS-inhibitor that is capable of reversing the chronic-stress-induced increase in anxiety-like behavior [48]. On the other hand, the nNOS activity in the hippocampus induces a decrease in the expression of glucocorticoid receptors (GR) in the hypothalamus, thereby reducing the negative feedback that is induced by corticosterone [49]. Given this, the effects of the CUS and OUA treatment on the NOS activity in the hypothalamus and hippocampus were investigated, although no changes were observed (Figure 4). The participation of iNOS in chronic stress has been previously described [50]. This study’s results showed that CUS increased the iNOS activity in the hippocampus (Figure 4), which is in accordance with a previous study [47]. Therefore, CUS may interfere with immunity through a system overactivation, leading to low-grade inflammation, as previously shown by Munhoz and colleagues [17], where the 14-day CUS model exacerbated the activation of the factor NF-κB nuclear factor in the frontal cortex and hippocampus, induced by LPS via glucocorticoid secretion. Furthermore, OUA has been related to its anti-inflammatory potential in several models [28,38], and in the present study, it was demonstrated that OUA treatment reduced the iNOS activity in the animals that were submitted to the CUS protocol (Figure 4).

Chronic stress may increase the ROS levels, as well as reduce the activity of antioxidant enzymes such as glutathione (GSH) and superoxide dismutase (SOD) [23,51]. The imbalance between free radical production and the body’s antioxidant capacity has been presented as an important factor in the development of neuropsychiatric diseases, including depression in humans and in rodents [52,53]. The present study demonstrates that the CUS group showed a reduction in the mRNA expression of the antioxidant enzyme, SOD2, in the hippocampus, which OUA was unable to revert (Figure 5). Interestingly, the treatment with OUA in the animals that were submitted to CUS reduced the mRNA expression of the glutathione-disulfide reductase gene (Gsr). This result suggests that OUA would avoid an increase in ROS production during CUS, thus necessitating more GSH transcription. This hypothesis is in accordance with a study that observed that OUA treatment reduced the oxidative stress in a rat hippocampus in a model of LPS-induced neuroinflammation [54].

Regarding cognitive performance, the present work shows that the rats that were treated with chronic intermittent OUA had an improvement in the long-term memory that was impaired by CUS (Figure 7). Interestingly, both the CUS paradigm and OUA treatment facilitated the rapid extinction of fear memory (Figure 8). Evidence from animal studies proposes that an acute moderate increase in corticosterone levels before exposure to a learning paradigm can enhance the memory consolidation of the new information, which includes the extinction memory. However, it can impair the retrieval of the information that is already stored [26]. Otherwise, in chronic stress or CUS, corticosterone impairs cognition; it reduces the magnitude of long-term potentiation (LTP) in hippocampal neurons, depressing the N-Methyl-D-Aspartate (NMDA) potentials that are triggered by glutamate [55]. This LTP impairment is detectable for two weeks after the last stressful event [56].

Indeed, our data demonstrated that an exposure to CUS was able to reduce the performance of rats in object recognition, 24 h after training, which is in agreement with previous studies [29]. Interestingly, the treatment with OUA reversed the long-term memory damage that was induced by CUS. The effect of OUA on long-term memory was previously reported by our group. When administered into the CA1 area of the hippocampus, OUA, at nanomolar concentration, increased the neuronal dendritic branching and improved the spatial long-term memory of healthy male adult rats. This was a long-lasting effect that was still observed after 14 days from the OUA injection [57]. OUA that was given to an animal, together with a protocol that is known to cause long-term memory impairment, enabled it to avoid this impairment. Some possible explanations point to glutamatergic signaling, which is closely related to Na^+^,K^+^-ATPase functions [58]. In cerebellar or hippocampal neurons, a treatment with OUA, in a non-inhibitory dose, led to NMDA activation [59,60]. Moreover, during synaptic activity, this NMDA activation was followed by an increase in Na^+^,K^+^-ATPase activity [61,62].

In order to assess context-dependent fear memory and extinction, the rats were exposed to a fear conditioning protocol 24 h after the last chronic unpredictable stressor stimulus, and analyzed for their recall and extinction of context-dependent fear for seven consecutive days. This fear memory consolidation shows a decreased efficacy with time after training and repeated presentations of the context, without an aversive stimulus causing a gradual reduction in the fear response, which is a form of learning that is referred to as fear extinction [63]. The results suggest that either the OUA or CUS promoted a faster extinction of fear memory, compared with the control group, without interfering with the acquisition of aversive memory (Figure 8).

It is well described in the scientific literature that the basolateral amygdala (BLA) is the structure that is involved in the development of conditional fear. However, evidence supports a role for this region in memory encoding, more than the storage of fear memory, without ruling out a time-limited role in memory consolidation [64]. Other studies have suggested that BLA can act synergistically with the hippocampus, which is responsible for the consolidation of context memory [65]. Thus, our results may suggest that OUA probably improves the new information or non-traumatic memory consolidation in the hippocampus, replacing the previous fear memory. However, more studies are necessary to elucidate the OUA mechanism, which is the mechanism that is triggered by CUS that leads to the increased fear extinction in this study, which should diverge from OUA.

The HPA axis activity, in response to the stressors stimuli and contextual fear memory, depends on a neuronal circuitry that involves the hippocampus, medial prefrontal cortex, and amygdala, as well as the participation of glucocorticoid and the CRH neuropeptide [66,67]. Recent studies point to the participation of CRHR1 in conditioned fear memory, and it has been shown that antalarmin, a CRHR1 antagonist, when administered systemically, attenuated the fear response, as well as rescued the HPA axis activity in rats [68]. In the present study, a decrease in the mRNA levels of the Crhr1 in the hippocampus was observed in the group that received the OUA and was submitted to CUS.

In addition, the duration of the extinction training reflects the number of shocks that were used during the contextual fear conditioning. A higher numbers of shocks generated stronger fear associations, which required more time to with this extinction training [69].

Finally, the CUS regiment is also susceptible to decreased physiological responses over time, and this “stress habituation” state does not mean a return to the normal physiologic status, but long-term changes in the brain, which include the up-regulation of the CRH expression in the paraventricular nucleus of the hypothalamus [70]. Our results proposed that the CUS protocol could induce a rise in the levels of the Crh mRNA in the hypothalamus. OUA reverted it to basal levels (Figure 6). Based on this, our hypothesis is that, for the group submitted to the CUS regiment, just a 1 s foot shock represented a routine stressful section, which was more easily extinguishable in the CUS group because they were adapted to daily stressful sections.

## 5. Conclusions

In conclusion, these findings suggest a novel role for OUA-Na^+^,K^+^-ATPase, as an important regulator in the mechanisms that are triggered by CUS in the CNS, protecting rats from high corticosterone levels, changes in the HPA, the mediators involved in neuroprotection, and the long-term memory impairment induced by CUS, and in helping non-stressed rats to forget a traumatic memory (Figure 9).

## Figures and Tables

**Figure 1 biomedicines-11-01177-f001:**
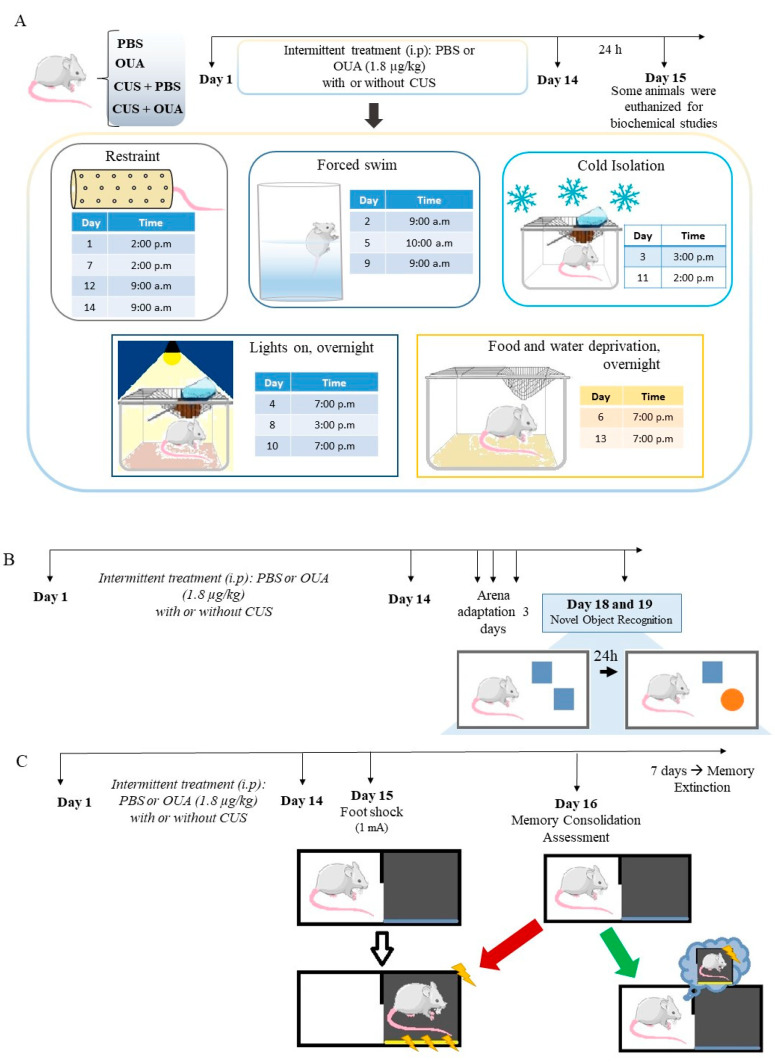
Schematic representation of the unpredictable chronic stress protocol and behavioral tests performed to assess the effects of CUS and intermittent OUA treatment on memory formation. (**A**) The animals were divided into four groups: PBS, OUA, CUS, and CUS + OUA. Intermittent treatment was performed, as early as day 1 the animals of the CUS and CUS + OUA group were exposed only to the stressor stimulus, and on day 2 the animals of the CUS and OUA + CUS were treated with PBS or ouabain 1 h before the stimulus stressor, so treatment was maintained alternately for 14 days. Control animals (PBS and OUA) were also treated on alternate days, but after treatment, they were returned to the cages. Twenty-four hours after the last stressor, the animals were euthanized, and the hippocampus and hypothalamus were stored for biochemical studies. (**B**,**C**) Different groups were exposed to behavioral testing 24 h after the last CUS protocol. (**B**) Scheme representative of the new object recognition test performed on the fourteenth day with the adaptation and 24 h after the test. (**C**) Illustrative scheme of contextual fear conditioning and memory extinction, first, the animals received a foot shock (1 mA) 24 h after the last stress stimulation. After 1 day, the animals were re-exposed to the arena, and the memory consolidation measure was performed. Subsequently, the extinction of fear memory was evaluated.

**Figure 2 biomedicines-11-01177-f002:**
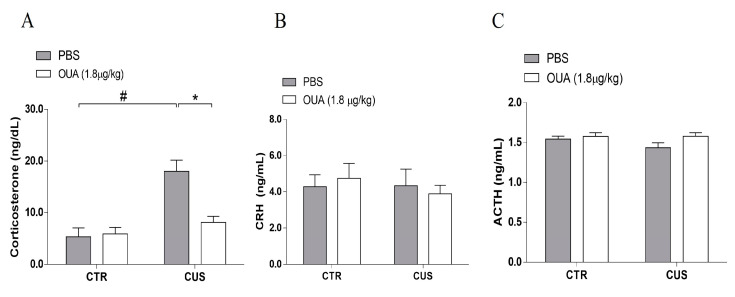
Ouabain reduces serum corticosterone levels in animals submitted to CUS. (**A**) Animals submitted to the CUS display an increase in serum corticosterone levels (ng/dL) (F (1, 24) = 0.58; *p* = 0.0001 for CUS factor), # *p* < 0.001 when compared to CTR x CUS; and * *p* < 0.001 when compared to CTR-CUS x CUS+OUA, 24 h after the last stressor stimulus, in relation to the CTR group (*n* = 7), and chronic treatment with OUA reduces corticosterone concentration in relation to the CUS group (F (1, 24) =8.108; *p* = 0.0089 for treatment factor). (**B**) CRH (ng/mL) (*n* = 13–15) (F (1, 51) = 3.117; *p* = 0.0835 for CUS factor. F (1, 51) =0.09387; *p* = 0.7606 for treatment factor). (**C**) ACTH (ng/mL) (*n* = 9–10) (F (1, 35) = 1.254; *p* = 0.2704 for CUS factor. F (1, 35) = 3.376; *p* = 0.0747 for treatment factor) had no alterations in the different groups studied, 24 h after last stressor stimulus. Data are presented as mean ± SEM (a two-way ANOVA, followed by Newman–Keuls post hoc test and a Kruskal–Wallis test, followed by Dun’s post hoc test, revealed a significance for corticosterone).

**Figure 3 biomedicines-11-01177-f003:**
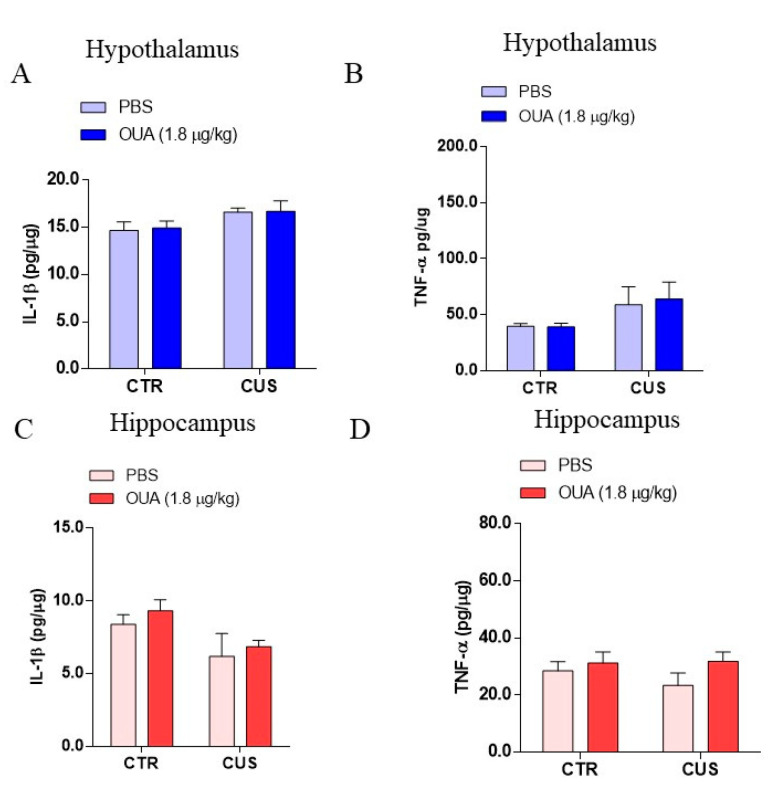
Effect of chronic unpredictable stress (CUS) on levels of pro-inflammatory cytokines in the hypothalamus and hippocampus. (**A**,**B**) Show levels of the hypothalamus (blue bars) (IL-1β (*n* = 5) (F (1, 16) = 4.914; *p* = 0.0415 for CUS factor. F (1, 16) = 0.03912; *p* = 0.8457 for treatment factor) and TNF-α (*n* = 9–10) (F (1, 34) = 4.338; *p* = 0.0449 for CUS factor. F (1, 34) = 0.04273; *p* = 0.8375 for treatment factor). (**C**,**D**) Levels of hippocampal (red bars) IL-1β (*n* = 5) (F (1, 16) = 6.002; *p* = 0.0262 CUS factor. F (1, 16) = 0.7027; *p* = 0.4142 for treatment factor) and TNF-α (*n* = 8–10) (F (1, 30) = 0.3623; *p* = 0.5518 for CUS factor. F (1, 30) = 2.289; *p* = 0.1408 for treatment factor). Data are presented as mean ± SEM (a two-way ANOVA, followed by Newman–Keuls post hoc test, revealed a significance for TNF-α from HP, or Kruskal–Wallis test, followed by Dunn’s post hoc test, for IL-1β from HT and HP).

**Figure 4 biomedicines-11-01177-f004:**
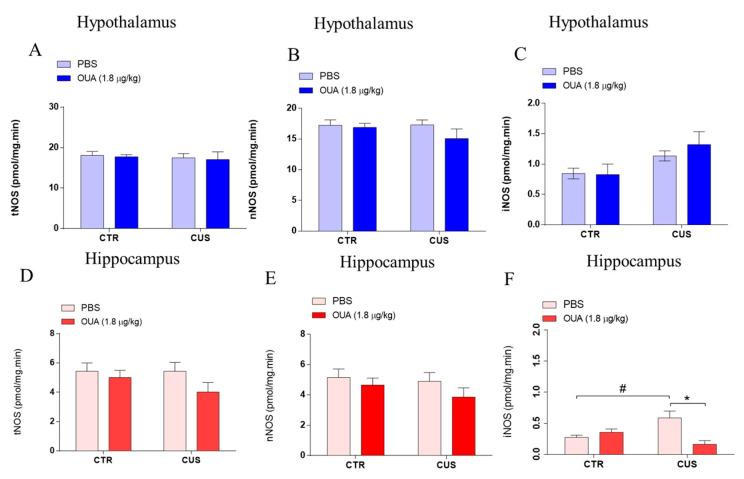
Chronic intermittent treatment with ouabain reduces iNOS activity in the hippocampus of animals submitted to CUS. (**A**–**C**) The activity of tNOS (F (1, 15) = 0.3099; *p* = 0.5860 for CUS factor. F (1, 15) = 0.1208; *p* = 0.7330 for treatment factor), nNOS (F (1, 16) = 0.6990; *p* = 0.4154 for CUS factor. F (1, 16) = 1.475; *p* = 0.2421 for treatment factor), and iNOS (F (1, 16) = 6.753; *p* = 0.0194 for CUS factor. F (1, 16) = 0.3043; *p* = 0.5888 for treatment factor) were not altered in the hypothalamus of the different groups studied (*n* = 5) (blue bars). (**D**,**E**) The graphs show that there was no change in NOS total (F (1, 16) = 0.7099; *p* = 0.4119 for CUS factor. F (1, 16) = 2.451; *p* = 0.1370 for treatment factor) and nNOS (F (1, 16) = 0.9158; *p* = 0.3528 for CUS factor. F (1, 16) = 1.991; *p* = 0.1774 for treatment factor) activity in the hippocampus of the different groups studied (red bars) (*n* = 5). (**F**) CUS increased iNOS activity, relative to control groups that were reduced by OUA treatment in the hippocampus (F (1, 16) = 0.6989; *p* = 0.4155 for CUS factor. F (1, 16) = 5.871; *p* = 0.0276 for treatment factor), # *p* < 0.05 when compared with CTR x CUS, and * *p* < 0.05 when compared with CTR-CUS x CUS+OUA (red bars) (*n* = 5). Results are presented as mean ± SEM (a two-way ANOVA, followed by Newman–Keuls post hoc test, revealed a significance).

**Figure 5 biomedicines-11-01177-f005:**
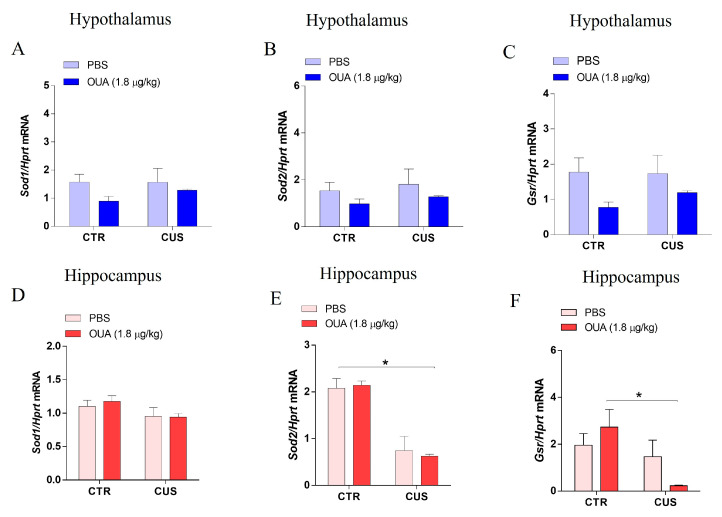
Modulation of antioxidant enzyme expression in rat hippocampus and hypothalamus by CUS protocol. (**A**–**C**) The results show no alteration in the expression of SOD1 (F (1, 34) = 0.4205; *p* = 0.5211 for CUS factor. F (1, 34) = 2.601; *p* = 0.1161 for treatment factor), SOD2 (F (1, 34) = 0.5226; *p* = 0.4747 for CUS factor. F (1, 34) = 1.922; *p* = 0.1746 for treatment factor), and GSR (F (1, 32) = 0.3697; *p* = 0.5475 for CUS factor. F (1, 32) = 6.277; *p* = 0.0175 for treatment factor) in the hypothalamus of the different groups studied (blue bars) (*n* = 8–10). (**D**) The results suggest an absence of modulation in SOD1 (F (1, 34) = 4.067; *p* = 0.0517 for CUS factor. F (1, 34) = 0.1041; *p* = 0.7489 for treatment factor) expression in the hippocampus of the different groups studied (*n* = 8–10). (**E**) SOD2 (F (1, 34) = 55.82; *p* < 0.0001 for CUS factor. F (1, 34) = 0.01967; *p* = 0.8893 for treatment factor), * *p* < 0.001, when compared with CTR x CUS expression, was reduced in the groups submitted to CUS in relation to the control groups in the hippocampus (*n*= 8–10). (**F**) GSR expression was reduced in the groups submitted to CUS and treated with OUA in relation to the control groups in the hippocampus (*n*= 8–10) (F (1, 30) = 7.688; *p* = 0.0095 for CUS factor. F (1, 30) = 0.1892; *p* = 0.6667 for treatment factor), * *p* < 0.05 when compared with CTR-OUA x CUS+OUA. Data are presented as mean ± SEM (a two-way ANOVA, followed by Newman–Keuls post hoc test, revealed a significance).

**Figure 6 biomedicines-11-01177-f006:**
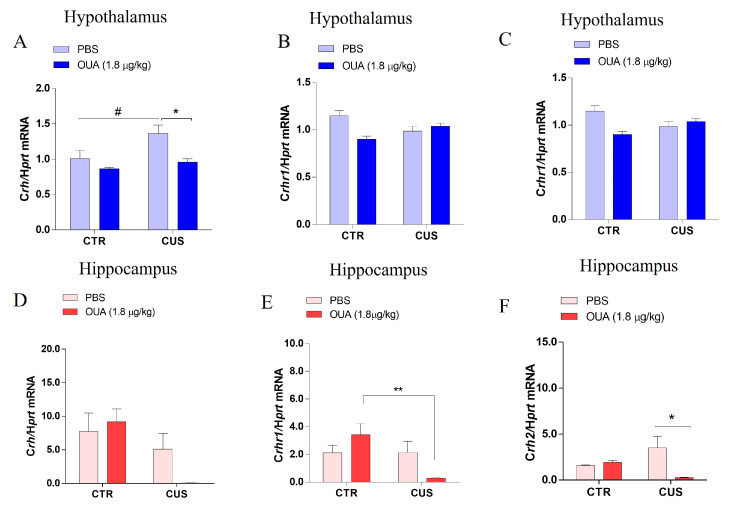
Decreased expression of Crh mRNA and their receptors in the hippocampus and hypothalamus of stressed rats and treated with ouabain. (**A**–**C**) Expression of Crh mRNA (*n*= 6–10) (F (1, 30) = 8.754; *p* = 0.0060 for CUS factor; F (1, 30) = 12.83; *p* = 0.0012 for treatment factor), # *p* < 0.05 when compared with CTR x CUS and * *p* < 0.05 when compared with CTR-CUS x CUS+OUA, Crhr1 mRNA (*n*= 8–10) (F (1, 34) = 0.08910; *p* = 0.7671 for CUS factor. F (1, 34) = 5.311; *p* = 0.0274 for treatment factor), and Crhr2 mRNA (*n*= 8–10) (F (1, 32) = 3.165; *p* = 0.0847 for CUS factor. F (1, 32) = 6.005; *p* = 0.0199 for treatment factor), * *p* < 0.05 when compared with CTR-OUA x CUS+OUA and # *p* < 0.05 when compared with CRT-CUS x CUS+OUA in the hypothalamus (blue bars). (**D**–**F**) Expression of Crh mRNA (*n*= 8–10) (F (1, 32) = 8.804; *p* = 0.0056 for CUS factor. F (1, 32) = 0.8223; *p* = 0.3713 for treatment factor), Crfr1 mRNA (*n*= 8–10) (F (1, 34) = 5.795; *p* = 0.0216 for CUS factor. F (1, 34) = 0.1837; *p* = 0.6709 for treatment factor), ** *p* < 0.01 when compared with CTR-OUA x CUS+OUA and Crhr2 mRNA (*n*= 8–10) (F (1, 30) = 0.03086; *p* = 0.8617 for CUS factor. F (1, 30) = 3.609; *p* = 0.0671 for treatment factor), * *p* < 0.05 when compared CTR-CUS x CUS+OUA in the hippocampus (red bars). Data are presented as mean ± SEM (a two-way ANOVA, followed by Newman–Keuls post hoc test, revealed a significance).

**Figure 7 biomedicines-11-01177-f007:**
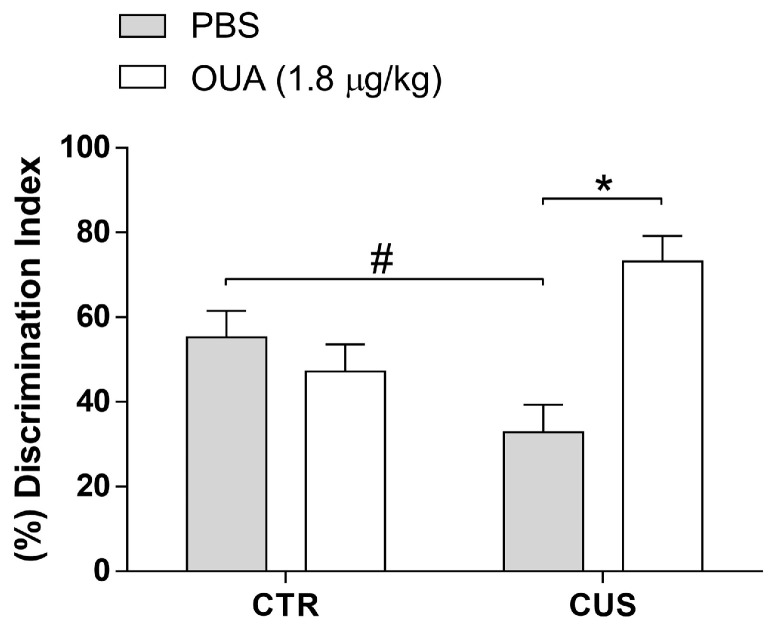
Ouabain (OUA) prevents chronic-unpredictable-stress-induced memory impairment. After 24 h of the last stressor stimulus, the animals were submitted to the test where they were exposed to two equal objects. After three days, an object was replaced, and the time of exploration of the two objects was quantified to evaluate the long-term memory. Data are represented by the percentage of the discrimination index (*n* = 7–9) (F (1, 26) = 0.08254; *p* = 0.7762 for CUS factor. F (1, 26) = 6.488; *p* = 0.0171 for treatment factor). Data are presented as mean ± SEM. # *p* < 0.05 when compared CTR-PBS x CUS, * *p* < 0.05 when compared with CTR-CUS x CUS-OUA (a two-way ANOVA, followed by Newman–Keuls post hoc test, revealed a significance).

**Figure 8 biomedicines-11-01177-f008:**
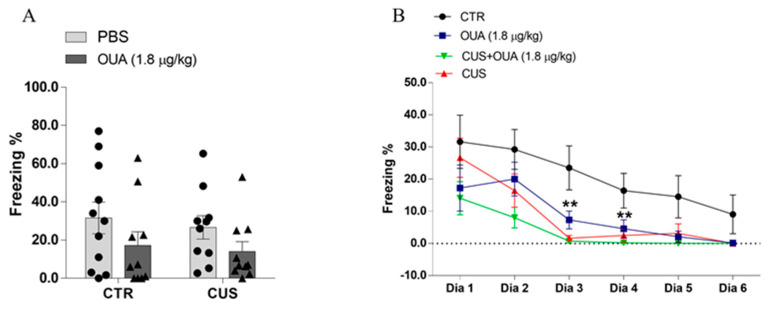
The effects of chronic ouabain administration and chronic unpredictable stress in the formation and extinction of fear memory. (**A**) Shows that there was no difference in the freezing percentage between the groups studied, 24 h after foot shock in the animals of the different groups (*n* = 10) (F (1, 37) = 0.3469; *p* = 0.5595 for CUS factor. F (1, 37) = 3.840; *p* = 0.0576 for treatment factor). (**B**) Animals from control group (CTR) (*n* = 10) presented higher percentage of freezing on days 3 (F (1, 37) = 12.94; *p* = 0.0009 for CUS factor. F (1, 37) = 4.663; *p* = 0.0374 for treatment factor) and 4 (F (1, 37) = 7.477; *p* = 0.0095 for CUS factor. F (1, 37) = 4.443; *p* = 0.0419 for treatment factor), compared to OUA (*n* = 10), CUS (*n* = 10), and CUS + OUA (*n* = 10) groups. Data are presented as mean ± SEM. ** *p* < 0.01 when compared with CTR-PBS x OUA; CTR-PBS x CUS; and CTR-PBS x CUS + OUA (t-student test was performed or a two-way ANOVA, followed by Newman–Keuls post hoc test, revealed a significance).

**Figure 9 biomedicines-11-01177-f009:**
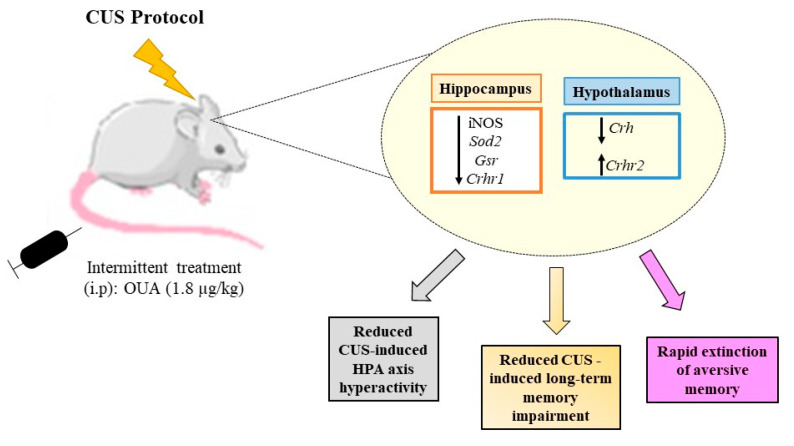
Schematic drawing of the proposed action upon OUA treatment in rats subjected to chronic unpredictable stress. Intermittent treatment with OUA reduced the activity of the HPA axis, since it reduced the expression of Crf and its Crfr1 receptor, leading to a reduction in the serum release of glucocorticoids, in addition, ouabain reduced activity of iNOS enzyme. The interference of treatment with OUA on the HPA axis promoted a rapid extinction of memory due to the fear of animals subjected to chronic stress.

**Table 1 biomedicines-11-01177-t001:** List of primers used for qPCR.

Gene	Primer ID
Crh	NM_031019.1
Crhr1	XM_006247542.2
Crhr2	NM_022714.2
Sod1	NM_017050.1
Sod2	NM_017051.2
Gsr	NM_053906.2
Hprt-1	NM_012583.2

## Data Availability

The data presented in this study are available upon request from the corresponding author.

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
