# Peer review of "Ouabain Reverts CUS-Induced Disruption of the HPA Axis and Avoids Long-Term Spatial Memory Deficits"

_biomedicines, 2023, doi:10.3390/biomedicines11041177_

Round 1

Reviewer 1 Report

The present work evaluated the effects of Ouabain on CUS-induced disruption of the HPA axis and spatial memory deficits in rats. The article is a very interesting and with a very good presentation.

I have some recommendations how to improve this article.

A severe correction of English language is needed. There are a lot of technical and grammatical mistakes in the whole text.

The section “Conclusion” is not fully understand. The clarification for all points of view is needed, especially for the application of plasma application from young females.

In conclusion, the article is not possible to publish at least at the present form.

Author Response

The present work evaluated the effects of Ouabain on CUS-induced disruption of the HPA axis and spatial memory deficits in rats. The article is a very interesting and with a very good presentation.

I have some recommendations how to improve this article.

A severe correction of English language is needed. There are a lot of technical and grammatical mistakes in the whole text.

R. The manuscript will be submitted to MDI English Edition Service before publication.

The section “Conclusion” is not fully understand. The clarification for all points of view is needed, especially for the application of plasma application from young females.

Replay: Thanks for your comment. However, in the study, measurements of Corticosterone, ACTH and CRH in the serum of male rats were used to evaluate the effects of Ouabain on the HPA axis of animals submitted to CUS. So, we reviewed the conclusion:

In conclusion, these findings suggest a novel role for OUA-Na+,K+-ATPase as an important regulator in the mechanisms triggered by CUS in CNS, protecting rats from high corticosterone levels, changes in HPA, mediators involved in neuroprotection and long-term memory impairment induced by CUS, and helping non-stressed rats to forget a traumatic memory. 

In conclusion, the article is not possible to publish at least at the present form.

Thank you for your time and effort in reviewing my research submission. Your thorough and insightful evaluations have been invaluable in helping me improve the quality of my work.

Reviewer 2 Report

In the present study, the authors examined the protective role of ouabain (OUA), a modulator of Na+, K+ -ATPase (NKA) activity, on rats suffered from chronic unpredictable stress (CUS). They investigated the effects of OUA (1.8 μg/kg) intermittent administration during CUS protocol in the rat’s central nervous system (CNS). They found that intermittent OUA treatment reversed CUS-induced HPA axis hyperactivity through the reduction of (1) 20 glucocorticoids levels, (2) CRH-CRHR1 expression, and by decreasing neuroinflammation with the reduction of iNOS activity, without interfering with the expression of antioxidant enzymes. Connectively, these findings suggest that a novel role for OUA-NPA interaction as an important regulator in the mechanisms triggered by CUS in CNS, protecting rats from the long-term memory impairment caused by stress, and helping non-stressed rats to forget a traumatic memory.

The manuscript was well written and the data are very interesting. I have some considerations for the studies.

1.  For all the figures, the authors described the significance or non significance for control group and CUS group, but it is better to mark it on bar graph in the figures.

2. In "Methods", the authors did not provide how many rats used in each group, please provide the numbers in your resubmission.

3. Line 50, it is not necessary to capitalized the first letter of the central nervous system.

4. Lines 153 to lines 156, there is repeated sentences "Learning and memory..... till test (5 min)". Please delete it. Also, please pay attention to it for your whole manuscript.

5. Lines 421 and 422, it is not necessary to use abbreviation for PTSD unless there is second time appears.

6. Line 530, please use the full name first followed by NKA if this is first appears.

7. The authors want to state their novelty for their study and many places used "for the first time" including abstract, discussion and conclusion. It is not necessary to repeat many times. For example, in "Conclusions", the authors stated " these findings suggest, for the first time, a novel role for .......". As you should know, the "novel" means no body reported before and similar meaning to "for the first time". Please carefully use " for the first time" in your manuscript. Better not to use this.

Author Response

Reviewer 2:

In the present study, the authors examined the protective role of ouabain (OUA), a modulator of Na+, K+ -ATPase (NKA) activity, on rats suffered from chronic unpredictable stress (CUS). They investigated the effects of OUA (1.8 μg/kg) intermittent administration during CUS protocol in the rat’s central nervous system (CNS). They found that intermittent OUA treatment reversed CUS-induced HPA axis hyperactivity through the reduction of (1) 20 glucocorticoids levels, (2) CRH-CRHR1 expression, and by decreasing neuroinflammation with the reduction of iNOS activity, without interfering with the expression of antioxidant enzymes. Connectively, these findings suggest that a novel role for OUA-NPA interaction as an important regulator in the mechanisms triggered by CUS in CNS, protecting rats from the long-term memory impairment caused by stress, and helping non-stressed rats to forget a traumatic memory.

The manuscript was well written and the data are very interesting. I have some considerations for the studies.

  1. For all the figures, the authors described the significance or non significance for control group and CUS group, but it is better to mark it on bar graph in the figures.

Reply: We kindly thank you for your observation. After analyzing the figures by the authors, we certified that all the significant differences are represented in the graphs with bars and no marks were add in the absence of significance. However, in order to maintain a clear description of the results all the statistical details were describe in the text.

  1. In "Methods", the authors did not provide how many rats used in each group, please provide the numbers in your resubmission.

Replay, thank you 212 rats were used to carry out all experiments. The total number of animals was added in the sentence: Male Wistar Rats (n=212; 53 per group) (250–350 g).

  1. Line 50, it is not necessary to capitalized the first letter of the central nervous system.

Reply: We appreciate the suggestion, we leave only the abbreviation for central nervous system, since it is written in line 33.

  1. Lines 153 to lines 156, there is repeated sentences "Learning and memory..... till test (5 min)". Please delete it. Also, please pay attention to it for your whole manuscript.

Reply: thank you, the duplicate sentence has been deleted

  1. Lines 421 and 422, it is not necessary to use abbreviation for PTSD unless there is second time appears.

Reply: thank you, we only indicated the abbreviation for post-traumatic stress, since the term appears in the introduction. “…parameters that are well known to be altered in depression, anxiety disorders and PTSD.”

  1. Line 530, please use the full name first followed by NKA if this is first appears.

Reply: thank you, the sentence was change to: Na+,K+-ATPase

  1. The authors want to state their novelty for their study and many places used "for the first time" including abstract, discussion and conclusion. It is not necessary to repeat many times. For example, in "Conclusions", the authors stated " these findings suggest, for the first time, a novel role for .......". As you should know, the "novel" means no body reported before and similar meaning to "for the first time". Please carefully use " for the first time" in your manuscript. Better not to use this.

Reply: We appreciate the suggestion, and the sentence has been removed throughout the text.

Abstract: “The present data demonstrate the ability of OUA to modulate the HPA axis as well as reverts CUS-induced long-term spatial memory deficits.”

Conclusion: “In conclusion, these findings suggest a novel role for OUA-Na+,K+-ATPase as an important regulator in the mechanisms triggered by CUS in CNS…”

 Thank you for your time and effort in reviewing my research submission. Your thorough and insightful evaluations have been invaluable in helping me improve the quality of my work.

Reviewer 3 Report

The work is interesting, even if there are several points to be clarified. Comments below

Introduction section

Pag. 1 line 33, please indicate full name for the CNS.

Page 2 line 47. I would suggest the authors rewrite this concept, emphasizing the rationale of this study. What is the purpose of this type of research? Understand the role of OUA or identify a molecular mechanism that is involved in stress conditions? In this regard, if the objective focuses on the endogenous Ouabain, why is the OAU taken into consideration? these clarifications are important in the introduction because they explain the real reason for this in vivo study.

Pag.2 line 50, please eliminate here full name for CNS:

Page 2 line 60: OUA reduces Crh mRNA levels…perhaps the authors mean Crh1mRNA?

Materials and Methods section.

Page 2 line 71: 

Ouabain (1.8 μg/kg): please report the manufacturer and report according to what criteria the choice of the concentration was made.

Page 4 line 103: ELISA test: 

Were all the measurements carried out at the same wavelength of 450 nm as reported in the paragraph? Also report in the text of the paragraph the units of measurement used to indicate the concentrations of the various cytokines and compounds tested in ELISA.

Page 4 line 110: 2.3. Measurement of NOS activity

The authors should be more precise in indicating which NOS activity it is, as there is no indication in the title of the paragraph and only in the middle of the speech does it emerge that it is iNOS.

Page 5: Par. 2.6. Contextual Fear Conditioning and Memory Extinction

On what basis were all these treatments conceived? Are they standardized protocols? Authors must provide justification.

Page 8. Fig. 4 

The authors report for INOS a significance of p<0.01 for iNOS when compared CTR-CUS x CUS+OUA. However, given that the columns of the graph show the mean ± SEM values, it is not very reliable that the attested difference is really significant. The authors should review the data from this statistical comparison of iNOS expression.

Page 10: Fig. 6

Panel F. Also in this case the p value *p < 0.05 when compared CTR-CUS x CUS+OUA in the hippocampus must be revised.

Page 11: Fig. 7

same observations on the indicated significance

Page 13 line 474,

What justification would the authors give to this transient effect of increasing inflammatory cytokines that they do not detect in their model? Given that the administrations are repeated and therefore chronic? This is an interesting aspect that should be better discussed.

Page 13 lines 486-494.

It is emblematic that only iNOS is regulated in a positive sense, being one of the main factors involved during inflammation. What justification do the authors give given that an inflammatory response cannot be deduced in these treated animals, as they themselves report, following the dosage of pro-inflammatory cytokines?

Page 15 Fig. 9: legend, line 578

I do not find it appropriate to talk about anti-inflammatory activity since the authors are based only on a reduction, however still not entirely clear, of iNOS expression, considering that there is no other evidence of anti-inflammatory actions in this study.

Author Response

Reviewer 3:

The work is interesting, even if there are several points to be clarified. Comments below

Introduction section.

Pag. 1 line 33, please indicate full name for the CNS.

Reply: thank you, the sentence was change to central nervous system.

Page 2 line 47. I would suggest the authors rewrite this concept, emphasizing the rationale of this study. What is the purpose of this type of research? Understand the role of OUA or identify a molecular mechanism that is involved in stress conditions? In this regard, if the objective focuses on the endogenous Ouabain, why is the OUA taken into consideration? these clarifications are important in the introduction because they explain the real reason for this in vivo study.

Reply: Thank you for your valuable suggestion.

Previous studies have shown that CUS promotes sensitization to neuroinflammation by immunological stimuli, such as bacterial lipopolysaccharides (LPS) (Munhoz et al., 2006) and knowing the anti-inflammatory effect of ouabain (Leite et al., 2022; Kinoshita et al., 2014; Leite et al., 2015), thus, the work aimed to investigate the molecular effects of treatment with OUA on the activity of the HPA axis, inflammatory and behavioral parameters in animals submitted to CUS.

Munhoz, C. D. Chronic Unpredictable Stress Exacerbates Lipopolysaccharide-Induced Activation of Nuclear Factor- B in the Frontal Cortex and Hippocampus via Glucocorticoid Secretion. Journal of Neuroscience 2006, 26 (14), 3813–3820. https://doi.org/10.1523/jneurosci.4398-05.2006.

Leite, J. A.; Cavalcante-Silva, L. H. A.; Ribeiro, M. R.; de Morais Lima, G.; Scavone, C.; Rodrigues-Mascarenhas, S. Neuroin-flammation and Neutrophils: Modulation by Ouabain. Frontiers in Pharmacology 2022, 13. https://doi.org/10.3389/fphar.2022.824907.

Kinoshita, P. F.; Yshii, L. M.; Vasconcelos, A. R.; Orellana, A. M. M.; Lima, L. de S.; Davel, A. P. C.; Rossoni, L. V.; Kawamoto, E. M.; Scavone, C. Signaling Function of Na,K-ATPase Induced by Ouabain against LPS as an Inflammation Model in Hippo-campus. Journal of Neuroinflammation 2014, 11 (1). https://doi.org/10.1186/s12974-014-0218-z.

Leite, J. A.; Alves, A. K. D. A.; Galvão, J. G. M.; Teixeira, M. P.; Cavalcante-Silva, L. H. A.; Scavone, C.; Morrot, A.; Rumjanek, V. M.; Rodrigues-Mascarenhas, S. Ouabain Modulates Zymosan-Induced Peritonitis in Mice. Mediators of Inflammation 2015, 2015, 1–12. https://doi.org/10.1155/2015/265798.

So, we made a change on line 47-55: “It has been suggested that endogenous Ouabain (OUA) levels are modulated by stress conditions [11], however little is known about the influence of this cardiosteroid in a presence of chronic unpredictable stress (CUS) situations. The plasma membrane protein Na+,K+-ATPase has the function of maintaining cellular ion homeostasis [12, 13]. Several CNS disorders are related to alterations in the Na+,K+-ATPase activity, such as depression and bipolar disorder [14-16]. Furthermore, knowing that CUS promotes neuroinflammatory sensitization [17] and OUA has anti-inflammatory and neuroprotective effects [18], in this work, we aimed to evaluate the role of intermittent treatment with OUA in chronic unpredictable stress (CUS)-induced HPA axis hyperactivity and its consequences in corticosterone levels, some inflammatory cytokines, antioxidant enzymes, as well as in long-term memory and in extinction of fear memory.

Pag.2 line 50, please eliminate here full name for CNS:

Reply: the sentence was deleted.

Page 2 line 60: OUA reduces Crh mRNA levels…perhaps the authors mean Crh1mRNA?

Reply: thank you, the sentence was change to: “Moreover, OUA reduces Crh mRNA levels in the hypothalamus, in addition to reducing Crh1 and Crh2 expression in the hippocampus.”

Materials and Methods section.

Page 2 line 71: 

  • Ouabain (1.8 μg/kg): please report the manufacturer and report according to what criteria the choice of the concentration was made.

Reply: we appreciate your question, the ouabain used was from Sigma-Aldrich. And the chosen dose of 1.8 μg/kg was due to our previous studies (Kinoshita, P. F.; Yshii, L. M.; Vasconcelos, A. R.; Orellana, A. M. M.; Lima, L. de S.; Davel, A. P. C.; Rossoni, L. V. ; Kawamoto, E. M.; Scavone, C. Signaling Function of Na,K-ATPase Induced by Ouabain against LPS as an Inflammation Model in Hippo-campus. Journal of Neuroinflammation 2014, 11 (1). https://doi.org/10.1186 /s12974-014-0218-z.).

The sentence was change to: or ouabain (1.8 μg/kg) [19] (Sigma-Aldrich).

Page 4 line 103: ELISA test: 

  • Were all the measurements carried out at the same wavelength of 450 nm as reported in the paragraph? Also report in the text of the paragraph the units of measurement used to indicate the concentrations of the various cytokines and compounds tested in ELISA.

Reply: thank you for your question, the measurable concentration ranges reported on the kits were indicated as requested. All kits were measured at 450 nm as required by the manufacturers of each kit.

We change the sentence to: ELISA kits measured serum levels of Corticosterone, CRH, ACTH following manu-facturer's instructions of Corticosterone EIA kit (32 - 20,000 pg/mL - Enzo Life Sciences In-ternational, Inc., USA), ACTH EIA kit (0.08 - 1.78 ng/mL- PHOENIX), and CRF EIA kit (0.33 - 3.73 ng/mL -PHOENIX). Hippocampal and Hypothalamus levels of TNF-α (39.1-2,500 pg/mL) and IL-1β (39-5,000 pg/mL) were measured by ELISA (eBioscience, USA) and the results were normalized by protein concentration. The absorbances were measured using a spectrophotometer at 450 nm (Epock, Bio-tech), and the concentrations of the cytokines were measured by correlating with the standard curve.

Page 4 line 110: 2.3. Measurement of NOS activity

  • The authors should be more precise in indicating which NOS activity it is, as there is no indication in the title of the paragraph and only in the middle of the speech does it emerge that it is iNOS.

Replay: we appreciate your observation; we have made a change in the explanation of the methodology.

2.3. Measurement of neuronal (nNOS) and inducible ( iNOS) NOS activity 

The constitutive NOS activity was measured by the [3H]L-citrulline assay method as previously described with some modifications [20]. This assay is based on the biochemical conversion of L-arginine to L-citrulline by NOS. For the NOS activity assay, the hypothalamus and hippocampus were homogenized in a buffer containing: 20 mM HEPES pH 7.4; 0.32 M sucrose; 0.1 mM EDTA; 1.0 mM DTT; 1.0 mM PMSF; 10 µg/mL leupeptin; 2 µg/mL aprotinin. Then, the tissue suspension was centrifuged at 1000 x g for 10 min, and the supernatants were centrifuged again at 12,000 x g for 20 min, 4 °C. After determining the protein concentration of the sample by the Bio-Rad kit, the samples were diluted to a concentration of 1 µg/µL, and samples were incubated for 30 minutes at 37 °C in 200 µL reaction medium containing: 20 µM arginine (0.5 μCi [3H]L-arginine), 4 µM FAD, 4 µM FMN, 10 µM BH4, 10 µg/mL Calmodulin, and 1mM NADPH and 100 µl sample. For iNOs activity (calcium/calmodulin-free activity) EDTA/EGTA were added and CaCl2 and calmodulin were omitted. The nNOS activity was calculated as the difference between the calcium-calmodulim sample and the EDTA/EGTA sample.To stop the reaction, the tubes were placed on ice, and 1 mL of 20 mM HEPES pH 5.5 was added. The total volume contained in the reaction portion was transferred to a column Dowex 50WX8-400, sodium form, to remove the excess of substrate. The resin was left to settle for 30 min at room temperature, the supernatant was care-fully removed in vials with scintillation liquid and the radioactivity to L-[3H] citrulline was quantified. Results were normalized by protein concentration and NO synthase activity was expressed as pmol/mg min. [20].

Page 5: Par. 2.6. Contextual Fear Conditioning and Memory Extinction

  • On what basis were all these treatments conceived? Are they standardized protocols? Authors must provide justification.

Reply: thank you for your question. The experiments were performed as described in: Novaes, L. S.; Bueno-de-Camargo, L. M.; Munhoz, C. D. Environmental Enrichment Prevents the Late Effect of Acute Stress-Induced Fear Extinction Deficit: The Role of Hippocampal AMPA-GluA1 Phosphorylation. Translational Psychiatry 2021, 11 (1). https://doi.org/10.1038/s41398-020-01140-6.The reference was added to the manuscript, number [22].

Page 8. Fig. 4 

  • The authors report for INOS a significance of p<0.01 for iNOS when compared CTR-CUS x CUS+OUA. However, given that the columns of the graph show the mean ± SEM values, it is not very reliable that the attested difference is significant. The authors should review the data from this statistical comparison of iNOS expression.

Replay: Thank you for your question. Follow the data obtained in the statistical test:

Newman-Keuls multiple comparisons test

Mean Diff,

Significant?

Summary

CUS:PBS vs. CUS:OUA

0,4240

Yes

**

Test details

Mean 1

Mean 2

Mean Diff,

SE of diff,

N1

N2

q

DF

CUS:PBS vs. CUS:OUA

0,5860

0,1620

0,4240

0,09980

5

5

6,008

16

(F) CUS increased iNOS activity relative to control groups that were reduced by OUA treatment in the hippocampus [F (1, 16) = 0,6989; P = 0,4155for CUS factor. F (1, 16) = 5,871; P = 0,0276 for treatment factor] #p <0.05 when compared CTR x CUS, *p < 0.05 when compared CTR-CUS x CUS+OUA

Page 10: Fig. 6

  • Panel F. Also in this case the p value *p < 0.05 when compared CTR-CUS x CUS+OUA in the hippocampus must be revised.

Replay: Thank you for your question. Follow the data obtained in the statistical test:

Newman-Keuls multiple comparisons test

Mean Diff,

Significant?

Summary

CUS:PBS vs. CUS:OUA

3,245

Yes

*

Test details

Mean 1

Mean 2

Mean Diff,

SE of diff,

N1

N2

q

DF

CUS:PBS vs. CUS:OUA

3,513

0,2676

3,245

0,9718

10

10

4,722

30

Page 11: Fig. 7

  • same observations on the indicated significance

Replay: Thank you for your question. Follow the data obtained in the statistical test:

Data are presented as mean ± SEM. #p <0.05 when compared CTR-PBS x CUS, *p < 0.05 when compared CTR-CUS x CUS-OUA (Two-way ANOVA followed by Newman-Keuls post hoc test revealed a significant).

Newman-Keuls multiple comparisons test

Mean Diff,

Significant?

Summary

CUS:PBS vs. CUS:OUA

-40,36

Yes

***

Test details

Mean 1

Mean 2

Mean Diff,

SE of diff,

N1

N2

q

DF

CUS:PBS vs. CUS:OUA

32,88

73,24

-40,36

8,702

7

9

6,559

26

Page 13 line 474,

  • What justification would the authors give to this transient effect of increasing inflammatory cytokines that they do not detect in their model? Given that the administrations are repeated and therefore chronic? This is an interesting aspect that should be better discussed.

Reply: we appreciate your question. Studies indicate that stressful stimuli increase pro-inflammatory cytokines for up to 6 hours after stressful stimuli (O'Connor et al., 2003). In addition, as shown in reference [45] (Wang et. al, 2017) cited in the manuscript in line 476, CUS alone is not able to increase the expression of pro-inflammatory cytokines. However, some studies indicate that CUS for 14 days promotes neuroinflammatory sensitization to peripheral immunological challenges showed the priming effects of CUS in neuroinflammatory events (Wang et al., 2017; Munhoz et. al, 2006).

O’Connor, K. A.; Johnson, J. D.; Hansen, M. K.; Wieseler Frank, J. L.; Maksimova, E.; Watkins, L. R.; Maier, S. F. Peripheral and Central Proinflammatory Cytokine Response to a Severe Acute Stressor. Brain Research 2003, 991 (1-2), 123–132. https://doi.org/10.1016/j.brainres.2003.08.006.

Wang, N.; Ma, H.; Li, Z.; Gao, Y.; Cao, X.; Jiang, Y.; Zhou, Y.; Liu, S. Chronic Unpredictable Stress Exacerbates Surgery-Induced Sickness Behavior and Neuroinflammatory Responses via Glucocorticoids Secretion in Adult Rats. PLOS ONE 2017, 12 (8), e0183077. https://doi.org/10.1371/journal.pone.0183077.

Munhoz, C. D. Chronic Unpredictable Stress Exacerbates Lipopolysaccharide-Induced Activation of Nuclear Factor- B in the Frontal Cortex and Hippocampus via Glucocorticoid Secretion. Journal of Neuroscience 2006, 26 (14), 3813–3820. https://doi.org/10.1523/jneurosci.4398-05.2006.

Sorrells, S. F.; Sapolsky, R. M. An Inflammatory Review of Glucocorticoid Actions in the CNS. Brain, Behavior, and Immunity 2007, 21 (3), 259–272. https://doi.org/10.1016/j.bbi.2006.11.006.

In this context, we change the sentence of line 474-480: “These findings agree with previous study that performed CUS for 14 days in rats, where CUS alone did not modulate the expression of pro-inflammatory cytokines [45]. Nonetheless, it was observed an increase of IL-1β and TNF-α in hippocampus after 1 hour of the last CUS protocol, in male mice subject to 14 days of CUS [46], suggesting that per-haps cytokines presented a peak immediately after stress.  However, some studies indicate priming effects of CUS in neuroinflammatory events to peripheral immunological challenges [17, 45].” 

Page 13 lines 486-494.

  • It is emblematic that only iNOS is regulated in a positive sense, being one of the main factors involved during inflammation. What justification do the authors give given that an inflammatory response cannot be deduced in these treated animals, as they themselves report, following the dosage of pro-inflammatory cytokines?

CUS for 14 days promotes neuroinflammatory sensitization to peripheral immunological challenges showed the priming effects of CUS in neuroinflammatory events (Wang et al., 2017; Munhoz et. al, 2006). Interestingly, some studies have shown that chronic stress models promote activation of pro-inflammatory pathways in the brain, particularly iNOS activity, that with sustained kinetics, resulting in neural damage (Madrigal et al. 2001; Harvey et al., 2004). Thus, we have hypotheses that, despite not observing changes in pro-inflammatory cytokines, our model imprinted a neuroinflammation that was reversed by treatment with ouabain.

Madrigal, J. L. M.; Moro, M. A.; Lizasoain, I.; Lorenzo, P.; Castrillo, A.; Boscá, L.; Leza, J. C. Inducible Nitric Oxide Synthase Expression in Brain Cortex after Acute Restraint Stress Is Regulated by Nuclear Factor ΚB-Mediated Mechanisms. Journal of Neurochemistry 2001, 76 (2), 532–538. https://doi.org/10.1046/j.1471-4159.2001.00108.x.

Harvey, B. H.; Stein, D. J.; Oosthuizen, F.; Brand, L.; Wegener, G. Stress?Restress Evokes Sustained INOS Activity and Altered GABA Levels and NMDA Receptors in Rat Hippocampus. Psychopharmacology 2003, -1 (1), 1–1. https://doi.org/10.1007/s00213-004-1836-4.

Page 15 Fig. 9: legend, line 578

  • I do not find it appropriate to talk about anti-inflammatory activity since the authors are based only on a reduction, however still not entirely clear, of iNOS expression, considering that there is no other evidence of anti-inflammatory actions in this study.

Replay: Thank you for your observation. We change the sentence to: “… in addition, ouabain reduced activity of iNOS enzyme.”

Thank you for your time and effort in reviewing my research submission. Your thorough and insightful evaluations have been invaluable in helping me improve the quality of my work.

Round 2

Reviewer 3 Report

No comments